

# Expression of *Aspergillus niger* glucose oxidase in *Pichia pastoris* and its antimicrobial activity against *Agrobacterium* and *Escherichia coli*

Yonggang Wang[1,2,*], Jiangqin Wang[1,2,*], Feifan Leng[1,2], Jianzhong Ma[1,2] and Alnoor Bagadi[1]

[1] School of Life Science and Engineering, Lanzhou University of Technology, Lanzhou, Gansu, China
[2] Key Laboratory of Drug Screening and Deep Processing for Traditional Chinese and Tibetan Medicine of Gansu Province, Lanzhou University of Technology, Lanzhou, Gansu, China
* These authors contributed equally to this work.

Corresponding authors
Yonggang Wang, wangyg@lut.cn
Jianzhong Ma, majz@lut.cn

## ABSTRACT

The gene encoding glucose oxidase from *Aspergillus niger* ZM-8 was cloned and transferred to *Pichia pastoris* GS115, a transgenic strain *P. pastoris* GS115-His-GOD constructed. The growth curve of *P. pastoris* GS115-His-GOD was consistent with that of *Pichia pastoris* GS115-pPIC9K under non-induced culture conditions. Under methanol induction conditions, the growth of the GOD-transgenic strain was significantly lowered than *P. pastoris* GS115-pPIC9K with the induced-culture time increase, and the optical densities of GOD-transgenic strain reached one-third of that of the *P. pastoris* GS115-pPIC9K at 51 h. The activity of glucose oxidase in the cell-free supernatant, the supernatant of cell lysate, and the precipitation of cell lysate was 14.3 U/mL, 18.2 U/mL and 0.48 U/mL, respectively. The specific activity of glucose oxidase was 8.3 U/mg, 6.52 U/mg and 0.73 U/mg, respectively. The concentration of hydrogen peroxide formed by glucose oxidase from supernatant of the fermentation medium, the supernatant of the cell lysate, and the precipitation of cell lysate catalyzing 0.2 M glucose was 14.3 µg/mL, 18.2 µg/mL, 0.48 µg/mL, respectively. The combination of different concentrations of glucose oxidase and glucose could significantly inhibit the growth of *Agrobacterium* and *Escherichia coli* in logarithmic phase. The filter article containing supernatant of the fermentation medium, supernatant of the cell lysate, and precipitation of cell lysate had no inhibitory effect on *Agrobacterium* and *E. coli*. The minimum inhibitory concentration of hydrogen peroxide on the plate culture of *Agrobacterium* and *E. coli* was $5.6 \times 10^3$ µg/mL and $6.0 \times 10^3$ µg/mL, respectively.

## INTRODUCTION

Since 1929, Fleming's discovery on bactericides prompted a search for antimicrobial substance in molds of the same genus. *Coulthard et al. (1942)* first described an antibacterial glucose aerohydrogenes (Notatin, firstly named as penicillin A) from

*Penicillium notatum* Westling. Almost at the same time, other substances including penicillin B and penatin also were isolated from *Penicillium* and exhibited good antimicrobial activity (*Bruggen et al., 1943*; *Kocholaty, 1943*; *Roberts et al., 1943*). Until 1963, Muller found that penicillin B and penatin were identical to notatin, and all of them belong to Glucose oxidase (GOD) (*Bentley, 1963*), which played the important roles in the inhibition of microbial growth. Thereafter, Glucose oxidase, referred to as an an ideal enzyme, has attracted the attention of researchers (*Bentley, 1963*; *Park et al., 2000*; *Crognale et al., 2006*; *Belyad, Karkhanei & Raheb, 2018*; *Li et al., 2019*; *Tu et al., 2019*).

Glucose oxidase (β-D-glucose: oxygen 1-oxidoreductase, GOD, EC 1.1.3.4) catalyzes the oxidation of glucose to gluconic acid and hydrogen peroxide in the presence of molecular oxygen according to the following reactions (*Hodgkins et al., 1993*; *Yamaguchi et al., 2007*; *Meng et al., 2014*):

$$\text{β-D-glucose} + O_2 \xrightarrow{\text{GOD}} \text{β-D-glucono-δ-lactone} + H_2O_2$$

$$\text{β-D-glucono-δ-lactone} + H_2O \xrightarrow{\text{Spontaneously}} \text{D-gluconicacid}$$

GODs are produced by molds such as *Aspergillus niger* and *Penicillium* (*Frederick et al., 1990*; *Meng et al., 2014*; *Qiu et al., 2016*; *Farshad, Amin & Catherine, 2018*). Many literatures reported that GOD could inhibit the growth of microbes in foods or food preparation media (*Tiina & Sandholm, 1989*; *Yoo & Rand, 1995*; *Li et al., 2019*). And it has been proven that this type of antibacterial compound is hydrogen peroxide ($H_2O_2$), which is active against $G^+$ and $G^-$ bacteria (*Malherbe et al., 2003*). This bacteriostatic effect of hydrogen peroxide is mainly attributed to the peroxidation of membrane lipids (*Roberts et al., 1943*; *Piard & Desmazeaud, 1991*). In laboratory-scale testing, refrigerated shelf life of GOD-treated fish was improved by 67% over untreated fish (*Field et al., 1986*). Moreover, GOD was able to inhibit growth of Pseudomonas spp. which are the main psychrotrophic spoilage microorganisms of chilled poultry (*Barnes & Impey, 1968*; *Cox et al., 1975*). GODs are also used in many medical applications. Sandholm and his co-workers suggested that all mastitis pathogens were sensitive to the glucose oxidase-lactoperoxidase system (*Sandholm et al., 1988*). GOD was also used as an antimicrobial agent in oral care (*Szynol et al., 2004*). The effect of honey on clearing infections in a wide range of wounds, which often did not respond to conventional therapy, was result of the antibacterial activity of hydrogen peroxide that is produced by GOD in honey (*Molan, 2001*; *Bang, Buntting & Molan, 2003*; *Khadivi et al., 2017*).

As above-mentioned descriptions, GODs are prepared mainly from the fermentation of *Aspergillus* (*Tu et al., 2019*), *Penicillium* (*Bodade, Khobragade & Arfeen, 2010*; *Khan et al., 2016*), *Bacillus* sp. (*Xu et al., 2018*), *Cladosporium neopsychrotolerans* (*Ge et al., 2020*), transgenic *Trichoderma reesei* (*Wu et al., 2017*), transgenic *P. pastoris* (*Park et al., 2000*; *Crognale et al., 2006*; *Yamaguchi et al., 2007*; *Rocha et al., 2010*; *Fang et al., 2015*; *Belyad, Karkhanei & Raheb, 2018*), and directly used in industry (*Wong, Wong & Chen, 2008*) as a control agent against pathogenic microorganism (*Hopkinsa et al., 2019*; *Lee et al., 2019*; *Li et al., 2019*) or a key catalyst for bioelectrochemical applications

(*Visvanathan et al., 2018*; *Mano, 2019*). However, very little information is available whether a glucose oxidase-secreting microbe could inhibit growth of its surrounding living things and become an ecological bacteriostatic agent. As we all know, chemical control is still the main method used to control the incidence of gray mold, chemical disinfectant could leave unsafe residues on plant materials and can drive resistance in pathogens, as well as contribute to environmental pollution. In this study, we here reported a new biological control strategy (*Dal et al., 2008*) for controlling the growth of pathogenic microorganism. Specifically, the gene coding for glucose oxidase from *A. niger* ZM-8 was expressed under the control of inducible alcohol oxidase 1 (AOX-1) promoter in yeast *P. pastoris*. The antimicrobial property of the glucose oxidase enzyme was evaluated.

## MATERIALS AND METHODS

### Strains and plasmids

A 1,749.0 bp GOD gene fragment was amplified from the genomic DNA of *Aspergillus niger* ZM-8 by the CTAB method (*Porebski, Bailey & Baum, 1997*). Primers for PCR were designed as Table S1 based on conserved sequences of glucose oxidase gene (No. JO5242) from GenBank Database, and then cloned into plasmid pUC19, The linearized vector pUC19-His-GOD by *Sma*I was inserted into the *S. cerevisiae* α-factor secretion signal molecule pPIC9k (Invitrogn, Carlsbad, CA, USA) to generate the expression vector pPIC9k-His-GOD under the action of the promoter AOX-1. The identified recombinant plasmid pPIC9K-His-GOD was linearized by *Bgl*-Π and transformed into *P. pastoris* GS115 cells by electroporation. The electro-competent *P. pastoris* GS115 cells were prepared using standard methods (*Manivasakam & Schiestl, 1993*). The electroporation was performed using a Gene Pulser (Bio-Rad, Hercules, CA, USA) at 1.5 kV, 40.0 μF, and 150.0 Ω according to manufacturer's instruction.

### Screening of clones and determination of biomass

The recombinant yeast clones were screened on yeast extract peptone dextrose (YPD) (1% (w/v) yeast extract, 2% (w/v) tryptone, 2% (w/v) dextrose, 2% (w/v) agar) plus 1 M sorbitol (YPDS) plates containing 100.0 μg/mL G418 (Invitrogen, Carlsbad, CA, USA) for 2.0–4.0 days. Potential high-level secretion transformants were obtained from the YPDS agar plates containing a higher G418 concentration (300.0 μg/mL). All these potential high-level secretion clones were confirmed by PCR using genomic DNA as the templates.

One colony was picked among several high copy clones obtained from the plate containing *P. pastoris* GS115-pPIC9k-His-GOD. *P. pastoris* GS115-pPIC9k was used as a negative control for the experiment. Clones were inoculated in Buffered Glycerol-complex medium (BMGY) (1% (w/v) yeast extract, 2% (w/v) tryptone, 100 mM Potassium Phosphate (pH 6.0), 1.34% (w/v) YNB, $4 \times 10^{-5}$ D-Biotin, 1% (w/v) glycerol), and cultured at 30 °C until $OD_{600} = 0.60$. The culture was then transferred to Buffered methanol-complex medium (BMMY) and cultured at 30 °C. The absorbance of growing culture was measured every 3 h.

## Expression of GOD in transgenic *P. pastoris* GS115

The *P. Pastoris* strains were cultivated in BMGY medium at 30 °C for 24 h. Biomass was generated after initial growth phase with glycerol as a carbon source. Finally, to induce AOX-1 dependent protein expression, the methanol fed-batch phase was started with methanol feed rate of 0.5 mL/12.0 h. Cell-free supernatant, the supernatant of cell lysate, and the precipitate of cell lysate was collected, and crushed by ultrasonic. The ultrasonication conditions used was 15 s, 25 s, 380 w, 99 times, and then stored at 4 °C (*Cereghino & Cregg, 2000*).

## Analysis of glucose oxidase activity

*Pichia pastoris* GS115-His-GOD-01 and *P. pastoris* GS115-pPIC9k were cultured at 30 °C with supplement of 0.5% methanol per 12 h. Activities of glucose oxidase from cell-free supernatant, cell lysate supernatant and precipitation were carried out according to *Gemba & Hara (1971)* with a slight modification. In detail, the reaction system consisting of 4.0 mL of reaction mixture (0.20 M Acetic acid-Sodium acetate, pH 5.2; 0.2 M glucose and 1.0 mM Indigo Carmine) and 2.0 mL of the appropriately diluted enzyme solution was added into and incubated for 10 min at 37 °C. Then, the reaction was stopped by boiling water bath for 13 min. The absorbance was measured at a wavelength of 615.0 nm. One unit (U) of Glucose oxidase activity was defined as the amount of enzyme, that can oxidize 1.0 μM of β-D-glucose to D-gluconic acid and $H_2O_2$ per 3 min at pH 6.0 and 30 °C. All reactions were performed in triplicate. The formula of enzyme activity as follows:

$$X_0 = [(A - A_0) \times K + C_0] \times V \times F$$

$X_0$: Enzyme activity; $A$: Absorbance value of trichloroacetic acid instead of glucose as the control; $A_0$: Absorbance value of the sample solution; $K$: Slope of the standard curve; $C_0$: Intercept of the standard curve. $V$: the volume of sample solution; $F$: the dilution factor.

## The specific activity of GOD

Protein concentrations of cell-free liquid, cell lysate supernatant and precipitation from *P. pastoris* GS115-His-GOD-01 and *P. pastoris* GS115-pPIC9K were determined by the method of Bradford (*Hammond & Kruger, 1988*). Absorbance was measured at 615 nm wavelength and specific activity of GOD was the value of activity divided by the value of protein concentrations.

## Antibacterial effects of glucose and glucose oxidase system on growth of *Agrobacterium* and *E. coli* in liquid medium

Glucose oxidase and glucose were used in three dilution-set combinations.
The concentration of glucose used were 1.0, 2.5 and 5.0 mg/mL respectively. The GOD was from fermentation supernatant of transgenic *P. pastoris* GS115-His-GOD-01 that was induced by methanol and fermentation supernatant from *P. pastoris* GS115-pPIC9K as control. Concentrations of GOD used was 1.0, 5.0 and 10.0 U/mL. The GOD and glucose solutions were added in the medium of YEP or LB and arranged in a Latin-square design to

study the effects of substrates and enzyme on growth of *Agrobacterium* LBA4404 and *E. coli* DH5α by measuring optical density in 600 nm.

## GOD antibacterial activity to *Agrobacterium* and *E. coli* on agar plates

The antibacterial activity of GOD produced by *P. pastoris* GS115-His-GOD-01 against *A. tumefaciens* LBA4404 and *E. coli* DH5α (stored in Dr. Ma Jianzhong's laboratory of Lanzhou University of Technology) was determined. Cultivate to bacterial liquid $OD_{600}$ = 1.0, spread on YPE (1% (w/v) yeast extract, 1% (w/v) tryptone, 0.5% (w/v) NaCl and 1.5% (w/v ) Agar) plates or LB (1% (w/v) yeast extract, 2% (w/v) tryptone, 2% (w/v) NaCl and 1.5% (w/v) agar) plates, on 51 h after adding methanol, add cell-free liquid, sonicate the supernatant by precipitating, collect the pellet and resuspend the pellet in an ice bath, immerse it in sterile filter paper, and inoculate with 0.20 M *A. tumefaciens* LBA4404 or *E. coli* DH5α on the surface of the glucose plate, observe its antibacterial effect.

## Antibacterial activity of hydrogen peroxide solution to *Agrobacterium* and *E. coli* on agar plates

The inhibitory effects of *A. tumefaciens* LBA4404 and *E. coli* DH5α and the minimum hydrogen peroxide concentration to inhibit bacterial growth were detected. *A. tumefaciens* LBA4404 was cultured with shaking at 28 °C to $OD_{600}$ = 1.0, and 200 µL was coated on YPE medium, and then put into filter paper with different concentrations of hydrogen peroxide solution, and cultured at 28 °C for 14 h, and observed antibacterial effect. *E. coli* DH5α was shake-cultured at 37 °C to $OD_{600}$ = 1.5, 200 µL was applied to LB medium, and then filter paper of different concentrations of hydrogen peroxide solution was placed. The culture was allowed to stand at 37 °C for 14 h to observe the inhibitory effect.

# RESULTS

## Vector construction and screening of transgenic *P. pastoris* clones

*Pichia pastoris* strain GS115 was transformed using linearized pPIC9K-His-GOD as described in materials and methods to yield *P. pastoris* GS115-His-GOD (Fig. S1). Twelve clones were obtained and confirmed by PCR-testing for gene integration. These clones were then screened on YPDS plates with different concentrations of Geneticin (G418), that is, 100 mM, 200 mM, and 300 mM, respectively. A positive transgenic clone, designated as *P. pastoris* GS115-His-GOD 01 was grown on the YPDS plate with a high Geneticin concentration and was considered for subsequent experiments.

## Expression of the GOD affecting the growth of the GOD-transgenic strain

Hydrogen peroxide, one of the products by GOD, injures living cells. In line with this statement, growth of the GOD-transgenic strain, *P. pastoris* GS115-His-GOD 01 was analyzed. Compared to *P. pastoris* GS115-pPIC9K, the growth of *P. pastoris* GS115-His-GOD 01 was slightly decreased during 51st h of incubation under GOD uninduced

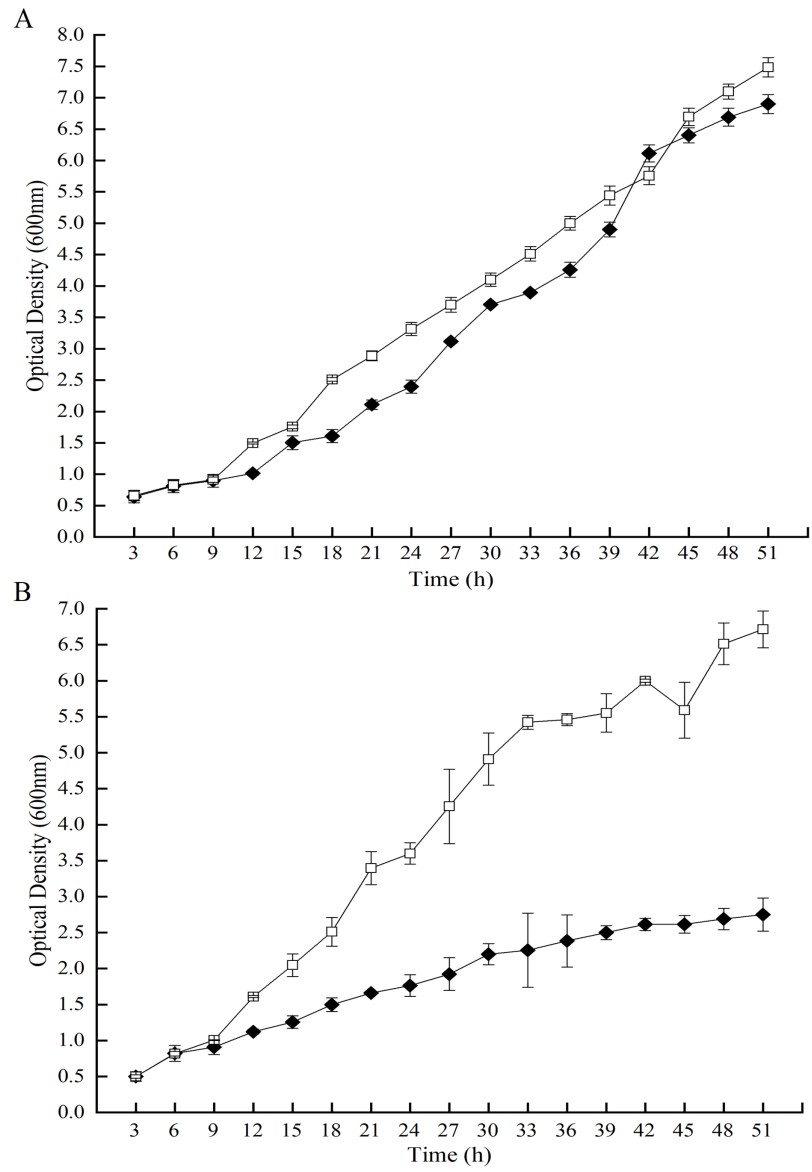

**Figure 1 Effects of GOD induction on the growth of the *P. pastoris* GS115-His-GOD 01.**
(A) *P. pastoris* GS115-His-GOD 01 (filled diamonds) and *P. pastoris* GS115-pPIC9K (hollow squares) were incubated in YPD without methanol for 51 h at 30 °C; (B) *P. pastoris* GS115-His-GOD 01 (filled diamonds) and *P. pastoris* GS115-pPIC9K (hollow squares) were incubated in YPD with 0.5% methanol added every 12 h for 51 h at 30 °C.

condition (Fig. 1A). Its optical density at 600 nm was 0.95-fold of that of *P. pastoris* GS115-pPIC9K at the time point of 51.0 h. However, the growth of *P. pastoris* GS115-His-GOD 01 was significantly lowered if the GOD was induced by methanol (Fig. 1B). During the growth of 51.0 h, the optical densities of *P. pastoris* GS115-His-GOD 01 were 0.54-fold of that of *P. pastoris* GS115-pPIC9K at 18.0h, 0.43-fold at 36 h, and 0.37-fold at 51.0 h, respectively. The inhibited growth of the GOD-transgenic *P. pastoris* could be attributed to the expression of the foreign GOD and, hereafter, accumulation of $H_2O_2$.

## Activities of the glucose oxidase

After 51 h-induced incubation, the cultures were processed into three parts of which were the cell-free supernatant, the supernatant and the precipitation of the cell lysates. The activities of the GOD preparations from *P. pastoris* GS115-His-GOD 01 were 14.27 U/mL in the cell-free supernatant, 18.2 U/mL in the supernatant of the cell lysate, and 0.48 U/mL in the precipitation (Fig. 2A). As a control, the activities of the three GOD preparations from *P. pastoris* GS115-pPIC9K were 3.22 U/mL, 1.76 U/mL and 0.41 U/mL, respectively (Fig. 2A). The specific activities of the three GOD preparations from *P. pastoris* GS115-His-GOD 01 were 8.30 U/mg in the cell-free supernatant, 6.52 U/mg in the supernatant of the cell lysate, and 0.73 U/mg in the precipitation, respectively (Fig. 2B). The specific activities of the three preparations from *P. pastoris* GS115-pPIC9K were 0.859 U/mg, 1.483 U/mg, and 0.529 U/mg, respectively (Fig. 2B). According to the specific activities, the cell-free supernatant of *P. pastoris* GS115-His-GOD 01 had the highest value, but the supernatant of the cell lysate of *P. pastoris* GS115-pPIC9K gave the highest specific activity. These results suggested that the native GOD of *P. pastoris* GS115 was mainly an intracellular enzyme. In the GOD-transgenic *P. pastoris* GS115, the enzyme was mainly secreted. This is in accordance with that the recombinant GOD was directed to an extra-cellular fraction by a signal peptide, α-mating factor.

## The concentration of hydrogen peroxide from GOD catalyzed glucose

The concentration of hydrogen peroxide produced by GOD from *P. pastoris* GS115-His-GOD-01 and *P. pastoris* GS115-pPIC9K catalytic glucose was showed in Fig. 3. According to the results in Fig. 3, the concentration of hydrogen peroxide was 14.3 μg/mL and 3.05 μg/mL in cell-free supernatant. The concentration of $H_2O_2$ from the supernatant of cell lysate was 18.2 μg/mL and 1.86 μg/mL, however, that of the precipitate of cell lysate was 0.48 μg/mL and 0.46 μg/mL. These results indicated that the GOD could be secreted out of the cells with the form of soluble protein.

## Inhibition of the GOD preparations on the growth of *A. tumefaciens* LBA4404 and *E. coli* in liquid medium

The GOD was prepared in fermentation supernatant of *P. pastoris* GS115-His-GOD 01. When the concentration of GOD was set as 1.0 U/mL, the growth curve of *A. tumefaciens* LBA4404 and *E. coli* DH5α under the gradually increasing of Glu were shown in Figs. 4A and 4B. From these two figures, almost no inhibition to the growth of *A. tumefaciens* LBA4404 compared with the control that was added with equal volume of *P. pastoris* GS115-pPIC9K fermentation supernatant. However, the marked inhibition to the growth of *E. coli* DH5α were observed from 4th h to 14th h with substrate concentration increasing (Fig. 4B). When the concentration of GOD was set as 5.0 U/mL, a slight inhibition on the growth of *A. tumefaciens* LBA4404 after 14th h with substrate concentration increasing (Fig. 4C), and the same change trends with Fig. 4B can be observed for the inhibition on *E. coli* DH5α from Fig. 4D. With the increasing of Glu concentration, the delay of growth of *A. tumefaciens* LBA4404 also be observed by the given GOD concentration of 10 U/mL (Fig. 4E), and the minor effect on *E. coli* DH5α also could be found in Fig. 4F. Conclusions

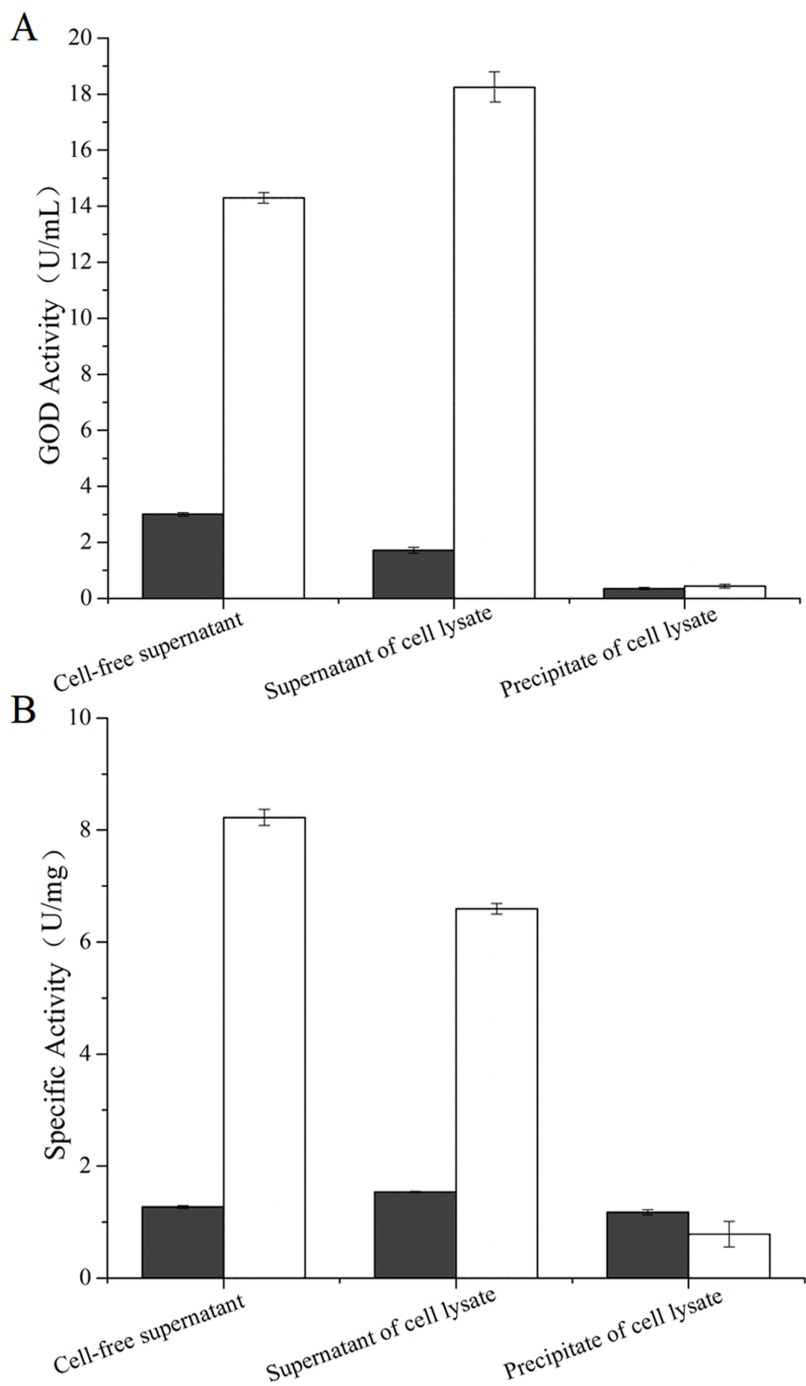

**Figure 2 Activities and specific activities of the GOD in the different preparations.** (A) The activities of the glucose oxidase in the cell-free supernatant, the supernatant of cell lysate, and the precipitate of cell lysate. (B) The specific activities of the glucose oxidase in the cell-free supernatant, the supernatant of cell lysate, and the precipitate of cell lysate. Data from *P. pastoris* GS115-pPIC9K are exhibited in black columns and *P. pastoris* GS115-His-GOD 01 in colorless columns.

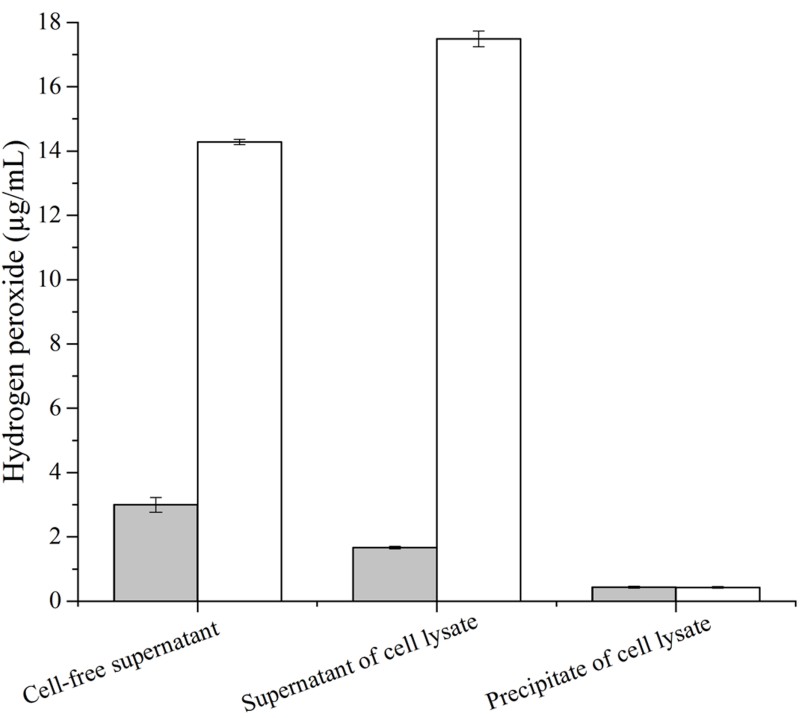

**Figure 3 Production of hydrogen peroxide by the GOD preparations.** Hydrogen peroxides produced by GOD from *P. pastoris* GS115-His-GOD 01 are in colorless columns, and in black columns from *P. pastoris* GS115-pPIC9K. Note: Cell-free supernatant, the GOD from the supernatant after fermentation; supernatant of cell lysate, the GOD from the supernatant of the cell lysate by ultrasonication; precipitate of cell lysate, the GOD from the precipitation of cell lysate by ultrasonication.

were drawn from Fig. 4, it showed these combinations did not completely inhibit growth of *A. tumefaciens* and LBA4404 *E. coli* DH5α, but influenced the time at which growth was initiated. Delay of growth initiation was greatest with the enzyme concentration, 5.0 U/mL, and the impact increased also with substrate concentration.

## Antibacterial effects of glucose and glucose oxidase on growth of *A. tumefaciens* and *E. coli* on agar plates

Analysis of the antibacterial activity of hydrogen peroxide ($H_2O_2$) produced by GOD catalyzed substrates glucose. *A. tumefaciens* LBA4404 (Fig. 5A) and *E. coli* DH5α (Fig. 5B) were plated on YPE or LB which were contained 0.2 M glucose. Filter papers were soaked by cell-free supernatant, the supernatant of cell lysate, and the precipitate of cell lysate from *P. pastoris* GS115-His-GOD 01, cell-free supernatant of *P. pastoris* GS115-pPIC9K as the negative control. The results showed that $H_2O_2$ derived from glucose which catalyzed by GOD had no effect on the growth of *A. tumefaciens* LBA4404 and *E. coli* DH5α.

## Antibacterial activity of hydrogen peroxide solution to *A. tumefaciens* LBA4404 and *E. coli* DH5α

To detect the minimum concentration of hydrogen peroxide solution inhibit the growth of *A. tumefaciens* LBA4404 and *E. coli* DH5α, the sterile filter articles were soaked with a

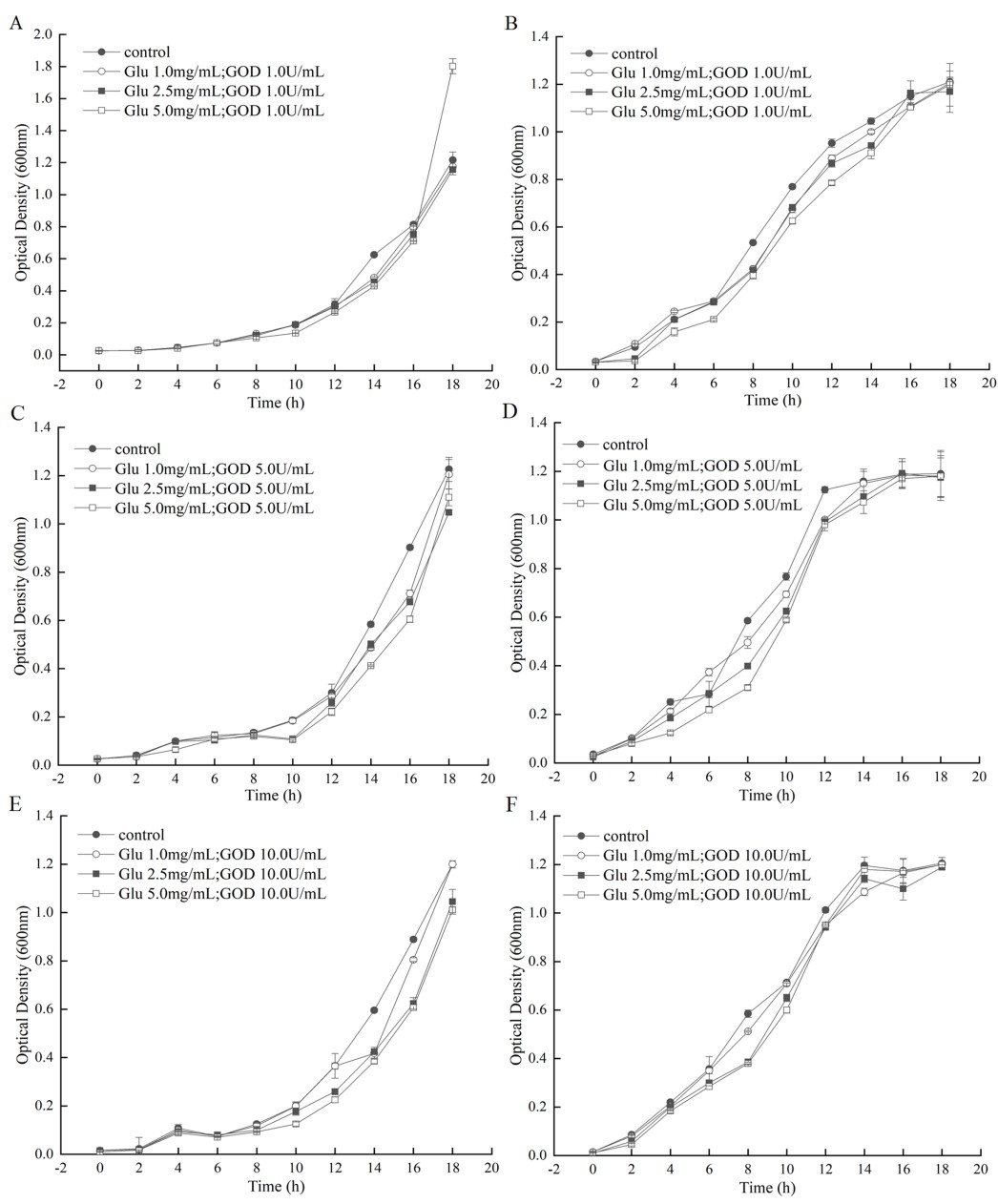

**Figure 4 Inhibition effects of the GOD preparations on growth of *A. tumefaciens* LBA4404 and *E. coli* DH5α in liquid media.** (A and B) Shows the inhibition effects of the GOD (1.0 U/mL) on growth of *A. tumefaciens* LBA4404 and *E. coli* DH5α with the final concentrations of glucose were 1.0 mg/mL, 2.5 mg/mL, 5.0 mg/mL, respectively. (C and D) Displayed the inhibition effects of the GOD (5.0 U/mL) on growth of *A. tumefaciens* LBA4404 and *E. coli* DH5α with the final concentrations of glucose were 1.0 mg/mL, 2.5 mg/mL, 5.0 mg/mL, respectively. (E and F) Displayed the inhibition effects of the GOD (10.0 U/mL) on growth of *A. tumefaciens* LBA4404 and *E. coli* DH5α with the final concentrations of glucose were 1.0 mg/mL, 2.5 mg/mL, 5.0 mg/mL, respectively.

volume of 10 μL hydrogen peroxide that was diluted to different concentrations. Different concentrations of hydrogen peroxide solution effect on *A. tumefaciens* LBA4404 and *E. coli* DH5α were determined. As shown in Figs. 6A–O and 6P–AA, the inhibition effect of

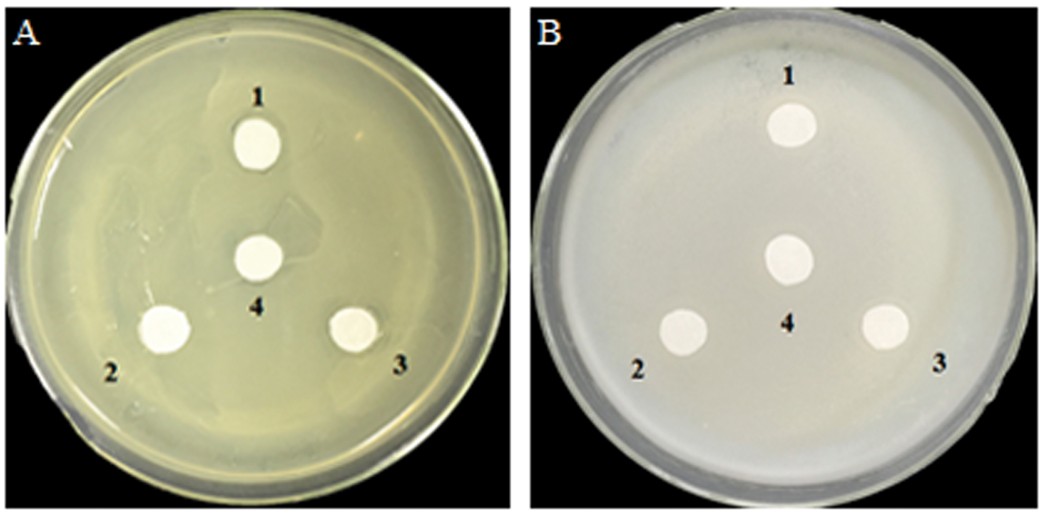

**Figure 5 Inhibition effects of the GOD preparations on growth of *A. tumefaciens* LBA4404 (A) and *E. coli* DH5α (B) in solid media.** (1) Denotes the GOD preparations from the cell-free supernatant of *P. pastoris* GS115-His-GOD 01; (2) is the supernatant of cell-lysate of *P. pastoris* GS115-His-GOD 01 by ultrasonication; (3) is the precipitate of cell-lysate of *P. pastoris* GS115-His-GOD 01 by ultrasonication; (4) represents the negative control with the sterile water.

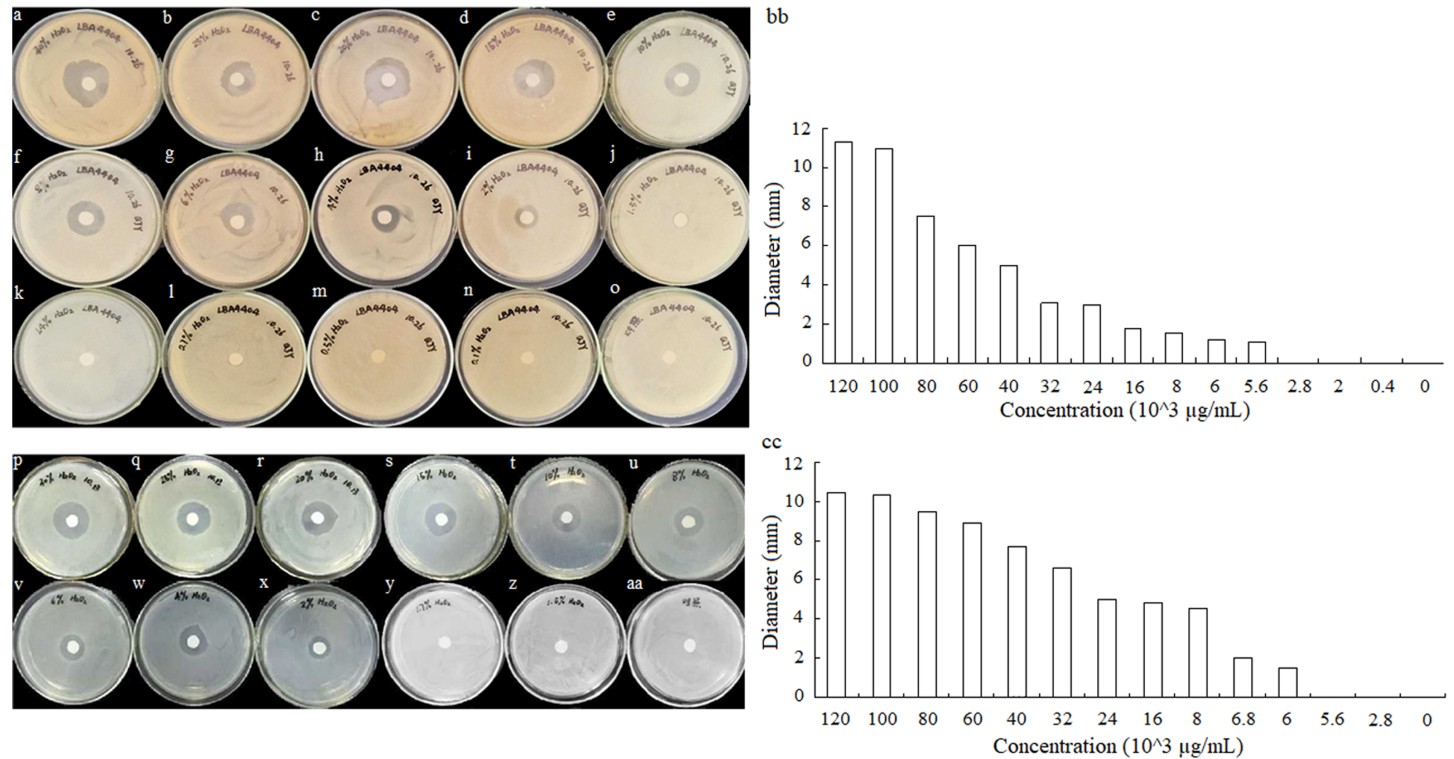

**Figure 6 Different concentrations of hydrogen peroxide solution effect on *A. tumefaciens* LBA4404 (A–O) and *E. coli* DH5α (P–AA) on agar plates.** The inhibit effects of different concentrations of hydrogen peroxide solutions on the growth of *A. tumefaciens* LBA4404 (BB), and *E. coli* DH5α (CC).
$H_2O_2$ on the two strains gradually decreased with the concentration reducing. The diameters of inhibition zone for these strains also were displayed in Figs. 6BB and 6CC, implying that the minimum concentration of hydrogen peroxide solution inhibits the growth of *A. tumefaciens* LBA4404 and *E. coli* DH5α was $5.6 \times 10^3$ μg/mL and $6.0 \times 10^3$ μg/mL.

## DISCUSSION

As evident from the abovementioned reviews, Glucose oxidase acts as a bacteriostatic agent by catalyzing hydrogen peroxide production via glucose oxidation (*Wong, Wong & Chen, 2008*; *Bankar et al., 2009*). At present, large-scale production of the enzyme was completed by good stability strains and fermentation technology (*Ge et al., 2020*). Although, many strains (*Bodade, Khobragade & Arfeen, 2010*; *Khan et al., 2016*; *Wu et al., 2017*) have been screened for producing this enzyme efficiently, only a few strains could be applied to commercial production. Therefore, more strategies need to be developed by constructing different bioreactor, modified enzyme, or enzyme engineering technology. Compared with a mass of glucose oxidase as an antibacterial agent applied in food preservation (*Lee et al., 2019*; *Li et al., 2019*), uses the GOD-transgenic strains or their fermented supernatants directly are easy, inexpensive, and widely available (*Dal et al., 2008*). However, little information is available whether a glucose oxidase-secreting microbe could inhibit the growth of its surrounding living things. As *Dal et al. (2008)* description, yeast could inhibit the gray mold growth, the novel viewpoint could be used to the development of bacteriostatic agent by genetic manipulations. Studies have found that most *P. pastoris* expression systems use methanol-induced ethanol oxidase promoters to express GOD (*Crognale et al., 2006*; *Yamaguchi et al., 2007*; *Belyad, Karkhanei & Raheb, 2018*), and the concentration of methanol directly affects cell growth and protein expression (*Cereghino et al., 2002*; *Xiao, Wang & Chen, 2004*; *Daly & Hearn, 2005*). *Crognale et al. (2006)* described a genetically modified *P. pastoris* X 33 with the gene encoding the GOX from *P. variabile* P16, and the activity only reached at 50 U/mL. *Kovačević et al. (2014)* cloned several mutated glucose oxidase genes from *A. niger* M12 and expressed them in *P. pastoris* KM71H. The highest activity of the GOD came up to 17.5 U/mL of fermentation media. *Gu et al. (2015)* reported recently that a yield of GOD reached 21.81 g/L, with an activity of 1972.9 U/mL, in *P. pastoris* S17 of which is a genetically modified strain by manipulating genes involved in protein folding machinery and abnormal folding stress responses. *Belyad, Karkhanei & Raheb (2018)* also cloned the GOD gene from *A. niger* ATCC 9029 and inserted into the pPIC9 vector for protein expression in *P. pastoris* GS115 by the alcohol oxidase promoter, but no the expression ability and enzyme activity were introduced. Although the above literatures has been reported the produce of GOD, there are no detailed determination on the activity of glucose oxidase in supernatant of the fermentation medium, the supernatant of the cell lysate, and the precipitation of cell lysate. In particular, there is no evidence of hydrogen peroxide production and bacteriostatic properties analysis. In this article, the GOD-encoding gene from *A. niger* ZM-8 was cloned and transferred into *P. pastoris* GS115 to yield a transgenic strain, which can excrete GOD to medium by the way of

methanol induction. The activity of glucose oxidase in supernatant of the fermentation medium, the supernatant of the cell lysate, and the precipitation of cell lysate was 14.3 U/mL, 18.2 U/mL and 0.48 U/mL, respectively. Corresponding these determined samples, The concentration of hydrogen peroxide formed by glucose oxidase can reached at 14.3 µg/ml, 18.2 µg/ml, 0.48 µg/ml, respectively. According to our results, the GOD-transgenic *P. pastoris* has to produce more enzyme molecules or higher active enzymes in order to inhibit microbes. Although the growth of *P. pastoris* GS115-His-GOD was found to be seriously inhibited during the period of methanol induction, its fermented supernatants containing the GOD activity can really reduce the growth of *E. coli* and *A. tumefaciens* in liquid culture (Fig. 4). In contrast, the GOD-soaked filter papers didn't exhibit any inhibition to the growth of *A. tumefaciens* and *E. coli* on the solid medium (Fig. 5). At present, it was not sure that it resulted from no enough oxygen or no enough GOD. As shown in Fig. 6, hydrogen peroxide can inhibit growth of *A. tumefaciens* and *E. coli* on solid medium, and the concentrations at least are $5.6 \times 10^3$ µg/mL and $6.0 \times 10^3$ µg/mL, respectively. To reach the concentration of hydrogen peroxide, the activity of the GOD produced from the transgenic strain should be at least increased 300-fold. To achieve antibacterial applications by GOD-transgenic *P. pastoris* directly, there will be more studies to be done in enzyme activity improvement and oxygen-offering system. These results could provide a new insight on bacteriostatic agent.

## CONCLUSION

This study cloned a gene encoding *Aspergillus niger* ZM-8 glucose oxidase and transferred it to *P. pastoris* to form a transgenic strain GS115-His-GOD. Compared with GS115-pPIC9K, the transgenic *P. pastoris* GS115-His-GOD could express glucose oxidase by methanol induction, The hydrogen peroxide could be produced with Glucose as the substrate during the fermentation process, and exhibited the inhibition activities on the growth of *A. tumefaciens* and *E. coli*.

### Funding

This study was financially supported by Chinese National Natural Science Foundation (Nos. 31760028 and 31460032), the Fundamental Research Funds for Key Laboratory of Drug Screening and Deep Processing for Traditional Chinese and Tibetan Medicine of Gansu Province (No. KZZY20180605), and the Youth Talent Support Program of Lanzhou University of Technology (No. 2018). The funders had no role in study design, data collection and analysis, decision to publish, or preparation of the manuscript.

### Grant Disclosures

The following grant information was disclosed by the authors:
Chinese National Natural Science Foundation: 31760028 and 31460032.
Key Laboratory of Drug Screening and Deep Processing for Traditional Chinese and

Tibetan Medicine: KZZY20180605.
Lanzhou University of Technology: 2018.

## Competing Interests

The authors declare that they have no competing interests.

## Author Contributions

- Yonggang Wang conceived and designed the experiments, performed the experiments, analyzed the data, prepared figures and/or tables, authored or reviewed drafts of the paper, and approved the final draft.
- Jiangqin Wang performed the experiments, analyzed the data, prepared figures and/or tables, authored or reviewed drafts of the paper, and approved the final draft.
- Feifan Leng performed the experiments, analyzed the data, prepared figures and/or tables, and approved the final draft.
- Jianzhong Ma conceived and designed the experiments, analyzed the data, authored or reviewed drafts of the paper, and approved the final draft.
- Alnoor Bagadi performed the experiments, analyzed the data, prepared figures and/or tables, and approved the final draft.

## Data Availability

The data is available in the figures and a Supplemental File.

## Supplemental Information

Supplemental information for this article can be found online at http://dx.doi.org/10.7717/peerj.9010#supplemental-information.

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
