# Peer review of "Expression of Aspergillus niger glucose oxidase in Pichia pastoris and its antimicrobial activity against Agrobacterium and Escherichia coli"

_PeerJ, doi:10.7717/peerj.9010_

## Round 0.1 · original submission · Major Revisions

Three specialists in the field evaluated the present manuscript, and they all have concerns related to this submission. The reviewers have described all points that should be answered by the authors. Please ensure that the English language in this submission meets our standards: uses clear and unambiguous text, is grammatically correct, and conforms to professional standards of courtesy and expression. Considering the evaluation carried out by all reviewers, I recommend major revision in this submission.

Reviewer 1 ·

Basic reporting

This paper explains about cloning and expression of the enzyme glucose oxidase in the methylotrophic yeast Pichia pastoris and its applications. The experiments have been well framed and executed. Inhibition of the growth of Agrobacterium and Escherichia coli by varying the concentration of glucose oxidase and glucose has been well explained. The article however misses to explain key points relevant to the study. The materials and method section, results and discussion section needs to rewritten. The structure of the article conforms to the PeerJ standards. Nevertheless, the article needs to be further improved and lacks clarity. Figures and raw data has been provided. References should be checked once again as per PeerJ standard.

Experimental design

The objective of this article falls within the scope of the journal. The research questions have been well defined, however the presentation of results and methods needs to improved.
Questions to the author:
1. The title may be modified as “Construction of Aspergillus niger glucose oxidase -transgenic Pichia pastoris and its antimicrobial properties”.
2. What was the growth rate of GS115 and GS115-His-GOD? It is better to mention the growth rate in the abstract.
3. The introduction part of the manuscript lacks earlier reports on GOD production in P. pastoris.
4. Line 52-55 needs can be rephrased as “In this study, the gene coding for glucose oxidase from Aspergillus niger ZM-8 was expressed under the control of inducible alcohol oxidase 1 (AOX1) promoter in yeast Pichia pastoris. The antimicrobial property of the glucose oxidase enzyme was studied”.
5. Line 57 – Title should be renamed as Strains and Plasmids
6. Line 61 should start with “The linearized vector was inserted in frame ……… promoter in pPIC9k (invitrogen) that resulted in expression vector pPIC9k-His-GOD”.
7. Line 66 – The electroporation was performed using a Gene pulser (Eppendorf) at 1.5 kV….”
8. Line 75-81 should be rephrased as “One colony was picked among several high copy clones obtained from the plate containing P. pastoris GS115-pPIC9k-His-GOD. GS115-pPIC9k was used as a negative control for the experiment. Clones were inoculated in Buffered Glycerol-complex medium (BMGY) [1% (w/v) yeast extract, ………. , glycerol] and cultured at 30°C until an OD….”. The culture was then transferred to Buffered methanol-complex medium (BMMY) and cultured at 30°C. The absorbance of growing culture was measured every 3 hours”.
9. Lines 83-89 should be rephrased as “The pichia strains were cultivated in BMGY medium at…. . Biomass was generated after an initial growth phase with glycerol as carbon source. Finally, to induce AOX 1 dependent protein expression, the methanol fed-batch phase was started with methanol ….. . ………… The ultrasonication conditions used was ………..”.
10. Lines 90 to 104 under analysis of glucose oxidase activity needs to completely rewritten.
11. Lines 106-109 should be rephrased as “Protein concentrations of cell-free liquid, ………were determined by Brandford method. Absorbance was measured at 615 nm ….”.
12. In Line 110 – Agrobacterium should be in italics
13. Line 112 to be rephrased as “The concentration of glucose used were 1, 2.5 and 5 mg/ml respectively. Line 115 – …………pPIC9k as control. Concentration of GOD used was 1, 5 amd 10 U/mL.
14. Maintain consistency for ml. It should be either ml or mL throughout the manuscript.
15. Rephrase lines 120 -136

Validity of the findings

1. Line 140 - ……. described in materials and methods section to yield …….
2. Line 143-144 – A positive transgenic clone, designated as P. pastoris GS115-His-GOD1 was grown on the YPDS plae with a high… and was considered for subsequent experiments.
3. Line 146 should be rephrased as “Hydrogen peroxide, one of the products produced by GOD, injures living cells. In line with this statement, growth of the GOD-transgenic strain, P. pastoris GS115-His-GOD 01 was analyzed”.
4. Line 147 should be rewritten as “Compared to P. pastoris GS115-pPIC9k ………. slightly decreased during 51st hour of incubation under GOD uninduced condition.
5. Line 177 – 184 should be rephrased and written clearly. Fig 4 title and its explanation should be clear. What is A to F should be clearly mentioned .
6. Did the author deduct the GOD activity of the control from the original value of the recombinant strain ?
7. If glucose oxidase was produced by the GS115 host itself, what was the rationale behind choosing glucose oxidase from another organism ?
8. Authors should try to explain in detail the novelty of the work in the discussion section. In what areas does this work standout from previous report on glucose oxidase and its antimicrobial properties? Compare with previous studies and discuss.
9. Quote latest references. Some of the references are too old.
10. Check references for PeerJ standards.
11. In figure 1 remove 3h, 6h, 9h, 12h etc. from x-axis. Instead just mention 3,6,9,12….
12. In figure 2, x-axis should be named as Cell-free supernatant, supernatant of cell lysate and precipitate of cell lysate. Activity of the supernatant of cell lysate is 18.2 U/ml but in fig. 2a it is around 16 U/ml only. Please check the bar graph. Error bars are missing in the bar graph.
13. Error bars or Standard deviation is missing in figure 3.

Additional comments

It is a good work and the quality of the article can be improved with clarity in the explanation. The english language should be improved to ensure that an international audience can clearly understand your article. Therefore, I would suggest the author to submit the article for English language correction.

Reviewer 2 ·

Basic reporting

The authors reported “ Construction of Aspergillus niger glucose oxidase transgenic Pichia pastoris and antimicrobial activity”
I regret to inform you that the work described in this manuscript does not greatly differ in properties from other glucose oxidases, and therefore the results are not very novel, although solid. All the text should be rewritten in good English. In fact, MS grammars and syntax should be revised with exact English usage.

Experimental design

No comment

Validity of the findings

No comment

Additional comments

No comment

·

Basic reporting

See general comments for the author

Experimental design

See general comments for the author

Validity of the findings

See general comments for the author

Additional comments

The manuscript " Construction of Aspergillus niger glucose oxidase-transgenic Pichia ‎pastoris and ‎antimicrobial activity" submitted by Wang et al. to publication in PeerJ ‎describes interesting data about the expression of Aspergillus niger glucose oxidase ‎in Pichia pastoris and especially antimicrobial activity of recombinant enzyme against ‎Agrobacterium and Escherichia.‎ But the manuscript is extremely poorly written and ‎needs to be largely reworked before considering for publication in PeerJ. I saw many ‎defects in terms of scientific writing and English grammar. Some sentences were ‎written confusing and difficult to understand. The manuscript is checked with the ‎help of an English language expert or service after that it resubmits for review ‎process. ‎
The Introduction must contain informative data about current work and not general ‎and well-document data. Authors have to review following references in Introduction ‎section: Aspergillus niger glucose oxidase was expressed in yeasts Saccharomyces ‎cerevisiae (Journal of Biological Chemistry, 1990; 265:3793–3802; Applied ‎Microbiology and Biotechnology, 2003; 61:502–511), Pichia pastoris (Protein ‎Expression and Purification, 2007; 55:273–278; Letters in Applied Microbiology, ‎‎2014; 58:393–400; Biotechnology and Biotechnological Equipment, 2016; 30:998–‎‎1005), Hansenula polymorpha (Yeast, 1993; 9:625–635), Kluyveromyces marxianus ‎‎(Microbial Cell Factories, 2010; 9:4) and Yarrowia lipolytica (Molcular Biotechnoloy, ‎‎2017; 59:307–314; World Journal of Microbiology and Biotechnology, 2018; 34:128). ‎Then they clearly mention their aims and novelty of current work at the end of the ‎Introduction section and also in the Abstract. ‎
I also recommend authors use the references for discuss their results in the ‎Discussion section. ‎

---

## Round 0.2 · accepted · Accept

The authors carried out all modifications indicated by the reviewers. In my view, the revised version of this manuscript can be accepted for publication as it is.